# PROCEEDINGS A

# Research

complexity, statistical physics, computer modelling and simulation

tipping point, complex network, sociophysics

**Authors for correspondence:**
Boris Podobnik
e-mail: bp@phy.hr
Petter Holme
e-mail: holme@cns.pi.titech.ac.jp
Tomislav Lipić
e-mail: Tomislav.Lipic@irb.hr

†These authors contributed equally to this study.

# The microdynamics shaping the relationship between democracy and corruption

Boris Podobnik[1,2,3,4,†], Marko Jusup[5,†],
Dean Korošak[6,7], Petter Holme[5] and Tomislav Lipić[8]

[1]Faculty of Civil Engineering, University of Rijeka, 51000 Rijeka, Croatia
[2]Faculty of Information Studies in Novo Mesto, 8000 Novo Mesto, Slovenia
[3]Luxembourg School of Business, 2453 Luxembourg, Luxembourg
[4]Zagreb School of Economics and Management, 10000 Zagreb, Croatia
[5]Tokyo Tech World Research Hub Initiative (WRHI), Institute of Innovative Research, Tokyo Institute of Technology, Tokyo 152-8552, Japan
[6]Institute of Physiology, Faculty of Medicine, and [7]Faculty of Civil Engineering, Transportation Engineering and Architecture, University of Maribor, 2000 Maribor, Slovenia
[8]Division of Electronics, Rudjer Boskovic Institute, 10000 Zagreb, Croatia

BP, 0000-0003-4270-1756; MJ, 0000-0002-0777-0425; PH, 0000-0003-2156-1096; TL, 0000-0002-8037-8198

Physics has a long tradition of laying rigorous quantitative foundations for social phenomena. Here, we up the ante for physics' forays into the territory of social sciences by (i) empirically documenting a tipping point in the relationship between democratic norms and corruption suppression, and then (ii) demonstrating how such a tipping point emerges from a micro-scale mechanistic model of spin dynamics in a complex network. Specifically, the tipping point in the relationship between democratic norms and corruption suppression is such that democratization has little effect on suppressing corruption below a critical threshold, but a large effect above the threshold.

The micro-scale model of spin dynamics underpins this phenomenon by reinterpreting spins in terms of unbiased (i.e. altruistic) and biased (i.e. parochial) other-regarding behaviour, as well as the corresponding voting preferences. Under weak democratic norms, dense social connections of parochialists enable coercing enough opportunist voters to vote in favour of perpetuating parochial in-group bias. Society may, however, strengthen democratic norms in a rapid turn of events during which opportunists adopt altruism and vote to subdue bias. The emerging model outcome at the societal scale thus mirrors the data, implying that democracy either perpetuates or suppresses corruption depending on the prevailing democratic norms.

## 1. Introduction

There is a long-standing interest in social phenomena among physicists [1–3], especially when it comes to the evolution of human cooperation [4–6]. Everyday life is fraught with dilemmas that require decisions between selfish interests and the common good. For the latter to prevail, people need to exhibit other-regarding behaviour. Such behaviour, if it comes at one's own expense, is referred to as altruism [7]. Cooperation is often said to be altruistic [8]. Interestingly, altruistic cooperation seems to defy Darwinian selection; why help others at a personal loss when ultimately it is the fittest who survive. It turns out, however, that there are multiple mechanisms [9] through which cooperators may recoup their losses in the long term [8], thus securing that selection favours cooperation after all. These mechanisms always involve a mutual recognition between cooperators [10], for instance kinship [11] or reputation [12].

Although mutual recognition between cooperators (also known as 'positive assortment' in the literature [10,13]) is considered essential for the evolution of cooperation, separating others into kin and non-kin or reputable and disreputable is not without its pitfalls. This is especially true when the separating trait gets conflated with one's own social group, implying that everyone in the group is kin or reputable and everyone on the outside is non-kin or disreputable. In such cases, altruism becomes biased towards the group and effectively turns into parochial in-group favouritism [14–17]. Cooperating within the group is then little more than the trade of favours because cooperators have considerable certainty that they will, over time, recuperate their losses with interest, as long as they stay loyal to the group. If, furthermore, favours are bestowed by abusing organizational properties and positions, as is often the case, favour trading is nothing but an individual-scale manifestation of corruption as a society-wide phenomenon [18].

Political economy argues that corruption is inescapable in social systems in which the over-concentration of power is sufficient to monopolize and allocate resources below market rates [18]. Democracy is seen as a means to prevent such power over-concentration, first because of election cycles that can oust corrupt officials [19] and second because of democratic pressures for transparency in government and business [20]. A large body of literature examines the relationship between democracy, corruption and economic growth with inconclusive results. Although earlier studies report that corruption may help firms avoid bureaucratic obstacles and thus generate growth [21], a more recent work in the same context expresses strong negative sentiments [22,23]. Similarly inconclusive results pertain to democracy and economic growth; again, earlier studies claim no relationship between political rights and growth [24], but a more recent work demonstrates that democracy has a definitive positive effect on growth in the long run [25]. Finally, when it comes to direct ties between democracy and corruption, an analysis of a measure of democracy consolidation and the World Bank's measure of corruption suggests an inverted-U relationship between the two over time [26]. This two-mode behaviour indicates that establishing democratic norms may initially lead to less control over corruption, but as the norms strengthen, so does the control. The underlying mechanism causing these two modes is still inadequately understood.

Here, a leap forward in understanding the democracy–corruption relationship is made, first by empirically analysing well-established indices of democracy and corruption to document a

surprising tipping point in how democracy suppresses corruption, and then by switching to mathematical modelling to unravel individual-scale forces that give rise to such a tipping point. In the process, we will have demonstrated that the democracy–corruption relationship is one of love and hate, that is, whether democracy perpetuates or suppresses corruption depends on the strength of the currently prevailing democratic norms.

## 2. Empirical analyses

Suspecting a non-trivial relationship between democracy and corruption, we examined the available data (figure 1). Specifically, we first used the Index of Democracy by the Economist Intelligence Unit as a proxy for the strength of democratic norms, and the Corruption Perceptions Index (CPI) by Transparency International as a proxy for corruption suppression (figure 1a). We found that, as might be expected, corruption suppression is positively associated with stronger democratic norms, but also that, below a certain threshold, the positive association is much weaker than that above the threshold. A statistical model of abrupt change nicely fits the data (figure 1a), showing a significant difference between the positive slopes before and after the threshold. To make sure that the described relationship is not just a peculiarity of the dataset, we repeated the analysis using completely independent proxies for the strength of democratic norms and corruption suppression: Control of Corruption by the World Bank and Global Freedom Scores by Freedom House, respectively (figure 1b). The same relationship emerges. Again, there is a threshold at which the positive association between corruption suppression and stronger democratic norms abruptly changes (figure 1b), thus confirming that our finding is not a data artefact, but rather a genuine feature of society. All data are available online for reanalysis (see Methods).

Quantitatively, the positive association between the Corruption Perception Index and the Index of Democracy has a critical threshold when the latter is at $\approx 6.77$, with the 95% confidence interval (6.30, 7.24). Below the threshold, suggesting a mode in which strengthening democratic norms marginally affect corruption suppression, the slope equals 3.04 (1.81, 4.26), but above the threshold, suggesting a mode in which the effect of strengthening democratic norms on corruption suppression is extensive, the slope is more than five times larger, equalling 16.84 (13.23, 20.44). Similarly for the other dataset, there is a threshold when Global Freedom Scores reach 83.0 (79.5, 86.5), while the gentle slope below the threshold at 0.0120 (0.0076, 0.0161) is an order of magnitude smaller than the steep slope at 0.132 (0.0980, 0.167). It would thus seem that democracy can be mostly powerless or extremely powerful against corruption, depending on whether the critical strength of democratic norms has been reached. This presents an intriguing situation whose mechanistic understanding may help policymakers and institutions to find more effective measures for fighting corruption.

Before constructing a mechanistic model, we also examined factors affecting voter turnout using the data on 50 democratic countries between 2009 and 2019 (figure 2a). The importance of this step was to determine whether there is an association between voter turnout and corruption suppression because keeping the status quo, such as rampant corruption, in a democracy is well served by having a large fraction of disheartened voters abstaining from voting. We included in the analysis a total of five factors, hypothesizing that voter turnout associates positively with (i) corruption suppression, (ii) democratic norms, (iii) gross domestic product (GDP) per capita, and (iv) membership of the European Union (EU), and negatively with (v) population size. We found that the data uphold hypotheses (i), (iii) and (v), but only the first two convincingly (figure 2b–e). It is surprising that voter turnout has no significant association with the strength of democratic norms (hypothesis (ii)), and even has a slightly negative association with membership of the EU (hypothesis (iv)). Most important for us, however, is that voter turnout associates positively with corruption suppression, suggesting that rampant corruption may indeed dishearten a large fraction of voters whose missing votes then potentially help perpetuate corruption.

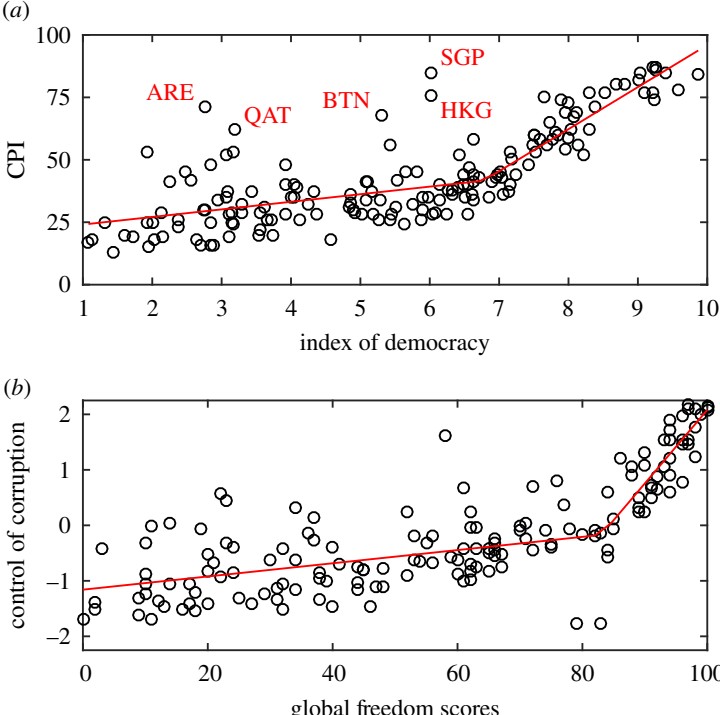

**Figure 1.** Two modes in the relationship between democracy and corruption. The panels show two independent indicators of the strength of democratic norms on the horizontal axes versus two independent indicators of control over corruption on the vertical axes. In both cases, democratization has little effect on suppressing corruption below a certain threshold, but a large effect above the threshold. The data are for 2019 (see also electronic supplementary material, figure S1). (*a*) Index of Democracy by the Economist Intelligence Unit versus the CPI by Transparency International. The statistics (with 95% confidence intervals) are: intercept 21.0 (15.4, 26.7), gentle slope 3.04 (1.81, 4.26), tipping point 6.77 (6.30, 7.24) and steep slope 16.8 (13.2, 20.4). Highlighted are the United Arab Emirates (ARE), Qatar (QAT), Bhutan (BTN), Hong Kong (HKG) and Singapore (SGP), which exhibit much better corruption suppression than the strength of democratic norms would suggest. All of these countries have relatively small populations, and either exceptionally high GDP per capita (ARE, QAT, HKG and SGP) or an exceptionally high GDP growth rate (BTN; 7.5% annually since the early 1980s). (*b*) Control of Corruption by the World Bank versus Global Freedom Scores by Freedom House. The statistics are: intercept −1.16 (−1.36, −0.95), gentle slope 0.0120 (0.0076, 0.0161), tipping point 83.0 (79.5, 86.5) and steep slope 0.132 (0.0980, 0.167). (Online version in colour.)

## 3. Mechanistic modelling

To better understand how it is possible that democracy and corruption coexist relatively peacefully in one societal mode, but oppose each other strongly in another, we resorted to mechanistic modelling at the scale of individuals. We started by embedding a set of $N$ agents into a social network such that agents are network nodes and friendships are links between nodes. Agents can be either (i) altruists, whose other-regarding behaviour is unbiased, (ii) parochialists, whose other-regarding behaviour is biased towards their own social group, or (iii) opportunists, who reside outside of the parochialist group but hope to join it. In the model, other-regarding behaviour manifests simply as friendships and thus is represented by links. The difference between agents is that altruists form, on average, $\langle k \rangle = k_0$ friendships indiscriminately, whereas parochialists form additional in-group friendships, $\langle k \rangle > k_0$, to facilitate the trade of favours and maintain disproportionate influence. Parochial in-group bias is thus seen as an individual-scale analogue of corruption. The dynamics of influence, inspired by a dynamical network model of

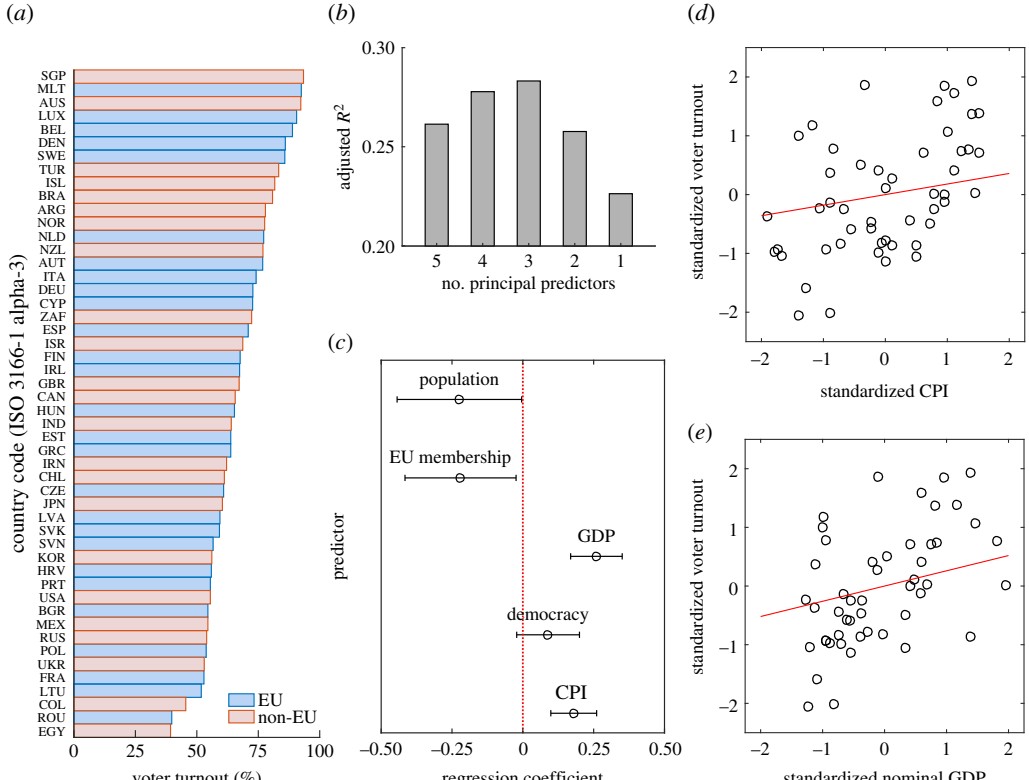

**Figure 2.** The average voter turnout increases with corruption suppression. Shown is, for a set of 50 countries between 2009 and 2019, the dependence of the average voter turnout on five socio-economic factors: corruption suppression (as measured by the CPI), strength of democratic norms (as measured by the Index of Democracy), nominal GDP per capita (as reported by the World Bank for 2019), membership of the EU and population size. We used principal component regression to disentangle the multicollinearity of socio-economic data (see Methods). (*a*) The average voter turnout between 2009 and 2019 in descending order. (*b*) The principal component regression yields the best results if three out of five principal predictors are retained (see Methods). (*c*) The average voter turnout increases with CPI and GDP, whereas EU membership and population have marginally significant negative effects. (*d*,*e*) The average voter turnout versus CPI and versus GDP visualized (see also electronic supplementary material, figure S2). (Online version in colour.)

generic node–node interactions [27] with applications in social contexts [28,29], is guided by the following three model assumptions (figure 3):

1. *Intrinsic preference.* An agent's intrinsic preference is encoded by a binary spin, $s_1$, with $s_1 = 0$ representing parochial and $s_1 = 1$ altruistic inclinations. We chose $s_1 = 0$ as the baseline, whereas change to $s_1 = 1$ occurs with a probability $p$ at any given time step and then persists for $\tau_1$ time steps.

2. *Induced preference.* Agents are responsive to social influence and peer pressure. We thus assumed that any agent $i$ with $k_i$ friendships harbours parochial inclinations if their neighbourhood comprises fewer than $m_u$ altruists, whereas a change is induced with probability $r$ at any given time step, persisting for $\tau_2$ time steps, if the neighbourhood comprises at least $m_u$ altruists [30]. These two alternatives are described by a second binary spin $s_2 = 0$ and $s_2 = 1$, respectively.

3. *Spread of parochialists.* Agents with spin $|s_1, s_2\rangle = |1, 1\rangle$ are altruists, comprising fraction $z$ of the population. Agents with spins $|1, 0\rangle$, $|0, 1\rangle$ and $|0, 0\rangle$ are all opportunists, comprising fraction $d$ of the population, until they are accepted into the parochial social group,

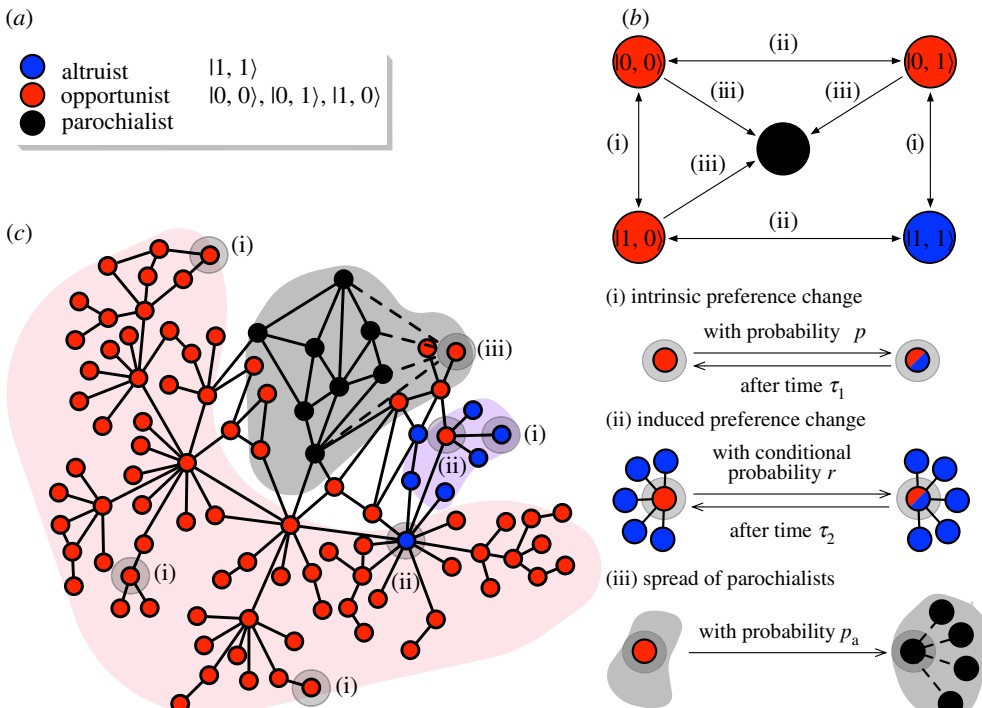

**Figure 3.** Microdynamics. The panels present a schematic overview of the model assumptions. (*a*) There are three agent types: altruists defined by state $|1, 1\rangle$; opportunists defined by states $|0, 0\rangle$, $|0, 1\rangle$ and $|1, 0\rangle$; and parochialists who are recruited from opportunists and are resistant to change owing to like-minded surroundings. (*b,c*) There are three types of processes through which opportunists can temporarily become altruists or permanently become parochialists: (i) intrinsic preference change that happens regardless of position in the social network; (ii) induced preference change that happens because of social influence and peer pressure; and (iii) parochialist spreading that happens when opportunists get accepted into the parochial social group forming new in-group connections. (Online version in colour.)

which comprises the remaining fraction $f$ of the population. Such acceptance occurs with probability $p_a$ at any given time step. In accordance with this assumption and the fact that $z + d + f = 1$, the fraction $f$ grows at a rate given by

$$\frac{\Delta f}{\Delta t} = p_a d = p_a(1 - z - f). \tag{3.1}$$

Once accepted into the parochial social group, agents form in-group friendships in excess of $k_0$. Of note is that, as the number of these excess friendships gets larger, parochial in-group bias in effect becomes an absorbing state; the spin $s_2 = 0$ occurs almost surely because of a like-minded neighbourhood.

To incorporate other relevant dynamic processes into the model, we made three additional assumptions:

4. *Evolution of altruism.* Parochial in-group bias is ancestral because it has served well ancient humans when between-group encounters spelled hostility rather than trade [31]. The circumstances have changed with the rise of civilizations, and especially modern-day globalization, bringing down barriers that have aided group inwardness in the past [32].

The probability $p$ is therefore subject to a slow positive drift

$$\frac{\Delta p}{\Delta t} = \epsilon(1-p),\tag{3.2}$$

where $\epsilon > 0$ is a small constant. The factor $1-p$ ensures that $p \le 1$.

5. *Voting.* Apart from parochialists, each agent is apathetic or enthusiastic about voting, which is described by a third binary spin $s_D = 0$ or $s_D = 1$, respectively. The baseline is $s_D = 1$, whereas change to $s_D = 0$ occurs with a probability $p_v$ at any given time step and then persists for T time steps. Parochialists are always enthusiastic about voting to preserve influence. The enthusiastic fraction of non-parochial agents according to this assumption is $\exp(-p_v T)$.

6. *Reforms.* If altruists win a majority, they instigate reforms that curb the spread of parochialists and promote enthusiasm for voting

$$\frac{\Delta p_\#}{\Delta t} = -\zeta_\# H[(z-d)\,e^{-p_v T} - f]p_\#,\tag{3.3}$$

where $p_\#$ stands for either $p_a$ or $p_v$, $\zeta_\#$ symbolizes reform rates $\zeta_a, \zeta_v > 0$, and $H[\cdot]$ is the Heaviside step function, guaranteeing that reforms happen only when altruists outvote parochialists and opportunists. The factor $p_\#$ on the right-hand side ensures that $p_a, p_v \ge 0$.

The influence dynamics implied by modelling assumptions 1–3 can be better understood via a mean-field approximation. For a randomly chosen non-parochial agent to exhibit altruistic intrinsic preference, its first binary spin must have flipped up, $s_1 = 1$, sometime over the past $\tau_1$ time steps, which happens with the above-mentioned probability $p^* = 1 - (1-p)^{\tau_1} \approx 1 - \exp(-p\tau_1)$. For the same agent to exhibit altruistic-induced preference two conditions must be satisfied. The agent's second binary spin must have flipped up, $s_2 = 1$, sometime over the past $\tau_2$ time steps, which happens with probability $r^* \approx 1 - \exp(-r\tau_2)$. Also, there must be at least $m_u$ altruists in this agent's neighbourhood, which is given by the probability

$$E = \sum_k \mathrm{zdf}(k) \sum_{j=0}^{m_u} \binom{k}{k-j} z^{k-j}(1-z)^j,\tag{3.4}$$

where $\mathrm{zdf}(k)$ is the fraction of agents with $k$ friendships. We therefore have

$$z = (1-f)p^* r^* E \tag{3.5}$$

and

$$d = (1-f)(1 - p^* r^* E).\tag{3.6}$$

From here, given the current fraction of parochialists $f$ in the population, we can calculate the fraction of altruists $z$ and the fraction of opportunists $d$. Interestingly, nonlinear equation (3.5) can yield more than one solution $0 \le z \le 1$, thus revealing a bi-stable nature of our model.

We confirmed the model's bi-stable nature using numerical simulations. The model's $(p^*, r^*)$ phase space splits into three distinct regions (figure 4a). For small $p^*$ and $r^*$ values, altruists are rare and opportunists are abundant. For large $p^*$ and $r^*$ values, the situation reverses, meaning that altruists are abundant and opportunists are rare. Finally, in the bi-stable region, both situations are possible; they even occur interchangeably under the right circumstances.

Before detailing the simulation results further, some helpful interpretations are in order. Assumption 1 implies that the probability $p^*$ quantifies society-wide intrinsic altruistic inclinations, whereas assumption 2 implies that the probability $r^*$ quantifies the society-wide strength of social influence and peer pressure. Owing to assumption 4, furthermore, the probability $p^*$ always increases in time, thus also serving as a simulation clock. Keeping these interpretations in mind, we ran simulations starting from the rare-altruist situation in which the $p^*$ value is small. With the passage of time altruism slowly evolves, the $p^*$ value increases and altruists gain some ground against opportunists. Such gradual betterment must, however, be

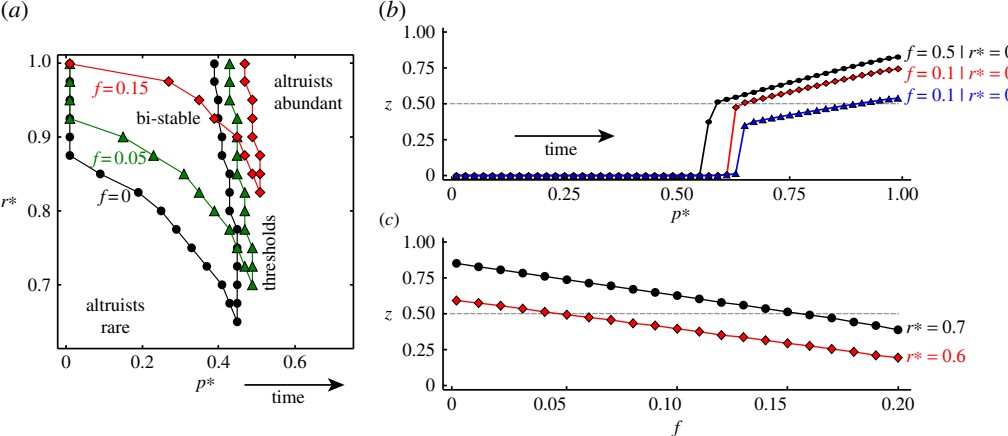

**Figure 4.** Society's path to an altruist majority. The panels demonstrate various aspects of the model dynamics. (*a*) The ($p^*$, $r^*$) plane of the model's phase space splits into a region with few altruists, a region with many altruists and a bi-stable region in which both situations are possible depending on the prevailing circumstances (see also electronic supplementary material, figure S3). As altruism slowly evolves over time (increasing $p^*$), society approaches a critical threshold that marks the border of the altruist-abundant region at which altruist abundance explodes. This border, however, shifts as the pool of parochialists expands (increasing $f$), thus becoming harder and harder to reach (see also electronic supplementary material, figure S4). (*b*) Equilibrium altruist abundance increases with the value of $p^*$, which in turn serves as a simulation clock. A key event is the explosive growth of altruists once the critical threshold is reached. The final altruist abundance depends on how big the pool of parochialists is allowed to get, $f$, and how strong social influence and peer pressure are, $r^*$. (*c*) Final altruist abundance decays with the size of the pool of parochialists. In particular, if this pool is allowed to get too big, altruist abundance may never be sufficient to win an election, thus postponing much needed reforms (see also electronic supplementary material, figure S5). (Online version in colour.)

accompanied by strong enough social influence and peer pressure, $r^* \gtrsim 0.6$, for a wider altruist expansion to take place. If this condition is satisfied, then there is a critical threshold (figure 4*a*) at which altruist abundance explodes (figure 4*b*), possibly even giving rise to an altruist majority.

Society's path to an altruist majority is by no means guaranteed. Rather it is a race. Owing to assumption 3, the pool of parochialists also increases over time, exerting a twofold effect on the population. First, the critical threshold shifts towards larger values of the probability $p^*$ (figure 4*a*), which delays the explosion of altruist abundance (figure 4*b*). Second, the critical threshold retracts towards ever larger values of the probability $r^*$ (figure 4*a*), which may prevent altruists from reaching the majority even after their abundance increases explosively (figure 4*b*). The explosive increase in the abundance of altruists may, in fact, become impossible altogether. For an altruist majority to ever happen, the speed at which altruism evolves relative to the expansion of parochialists must be high enough.

Even if society surpasses the critical threshold and altruists become abundant, they may still be unable to outvote parochialists and opportunists in an election. One reason for this is that the longer it takes for altruist abundance to explode, the larger is the pool of parochialists. Their dense social connections then limit the reach of altruists (figure 4*c*). Another reason is that parochialists are strongly motivated to vote in order to maintain their influence, whereas only a fraction of altruists (and similarly opportunists), $\exp(-p_v T)$, are enthusiastic about voting. Taking as an example that $f = 0.15$, it is visible from figure 4*c* that, even for a relatively large $r^* = 0.7$, the fraction of altruists is $z \approx 0.5$. In this situation, if enthusiasm for voting were as much as 80%, altruists would comprise $0.5 \times 0.8 \approx 40\%$ of the electorate, whereas parochialists and opportunists would comprise $0.15 + 0.35 \times 0.8 \approx 43\%$. Altruists would therefore fall short of winning the majority, which in turn would leave the influence of parochialists unscathed despite democratization.

# 4. Discussion

We have shown using two completely independent datasets that democracy and corruption are in a complicated relationship. Below a critical threshold, improving democratic norms has only a marginal effect on corruption suppression, but the effect is substantial above the threshold. We then employed mechanistic modelling to reveal the individual-scale origins of such a relationship. Dividing a population of agents into three types, altruists, parochialists and opportunists, we showed that parochial in-group bias can be used to maintain disproportionate influence and thus secure the vote needed to form a government whose agenda is avoiding reforms, especially those that would compromise parochialists' influence. Only if altruism evolved fast enough to trigger a societal change before parochialists entrench themselves too deep into the social fabric could altruists win the majority and form the government whose agenda is enacting reforms. Democracy can therefore both perpetuate and suppress parochial in-group bias, seen here as the individual-scale analogue of corruption.

In our introductory remarks, we briefly connected the biased other-regarding behaviour of parochial altruism as an individual-scale phenomenon and corruption as a society-wide phenomenon. Fortifying this connection, recent research shows that corruption can have roots in cooperation [33] and that humans direct their cooperativeness in-group rather than out-of-group [34] because they anticipate in-group cooperation to be reciprocated [34,35] even if groups are utterly random [35]. Analogous anticipation powers the evolution of cooperation in the general population through mechanisms [9] that offset cooperator costs [36,37], at least on average and in the long run. These mechanisms are, as the expressions 'on average' and 'in the long run' would suggest, intricate and intangible (e.g. reputation [38,39], social networks [40,41], etc.), whereas group membership is immediate and instinctive. A key consequence is that the act of in-group reciprocation, although still implicit, is much more certain than reciprocation in society at large. From the perspective of group members, out-of-group cooperation is risky, perhaps to the point of not being worth bothering with at all. If the exclusion is such that outsiders are denied valuable resources despite holding otherwise equal rights as group members, then in-group bias crosses the border of cronyism and, more generally, corruption [42].

Furthermore, we have shown empirically that the correlation between voter turnout and corruption suppression is positive. By including this effect in our mathematical model, we found how an intermediate-sized group of parochialists may cling to power. This can happen even if altruists are in the majority in the electorate and their enthusiasm for voting is relatively high. The problem lies in the abstention of apathetic voters that can be traced back to indifference and alienation [43], both of which have a non-negligible potential to affect election outcomes [43]. All this testifies to the importance of mobilizing the electorate, as well as predicting voter turnout. Appreciating the complexity of the latter problem [44], our analysis was by no means an exhaustive one. Instead, we aimed at exploring (and controlling for) factors that are of interest in the context of a general relationship between democracy and corruption. We thus found that the Index of Democracy is a poor predictor of voter turnout, which may seem surprising at first but fits naturally within our mathematical modelling framework. Only altruist voters mark the strengthening of democratic norms. It is, however, parochialists who are fervent voters, while opportunists also take part. Consequently, when altruists are apathetic about voting, we may observe weak democratic norms and decent voter turnout at the same time.

Our model mechanistically underpins the complicated relationship between democracy and corruption depicted in figure 1. The explosion of altruism beyond the critical threshold is a major societal event in whose wake altruists are likely to outvote parochialists and opportunists. Before this event, there are no incentives to coordinate democratization and corruption suppression (figure 5) because an influential social group (i.e. parochialists) benefits from cronyism and corruption. There is little reason for correlation between indices such as the CPI and the Index of Democracy. After the event, however, coordinating democratization and corruption suppression is essential (figure 5), precisely to prevent the 'old ways' from resurfacing. The aforementioned correlation should be strong, and the data confirm this.

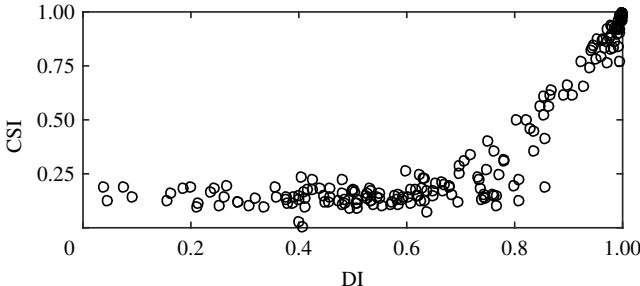

**Figure 5.** Model-generated relationship between paradigmatic democracy and corruption indices. The Index of Democracy (DI) on the x-axis measures how friendly a modelled society is to altruism *or* altruist voting in terms of the corresponding model parameters, that is, $\mathrm{DI} \propto p^* + \exp(-p_v T) - p^* \exp(-p_v T)$. The corruption (suppression) index on the y-axis measures how unfriendly the modelled society is to parochialist spreading, that is, $\mathrm{CSI} \propto 1 - p_a / \langle p_a \rangle$, where $\langle p_a \rangle$ is the average probability with which opportunists get accepted into the parochial social group before reforms. Shown are the results for a set of 200 simulated countries, with both indices arbitrarily re-scaled to values between 0 and 1.

We believe that the present study offers a fresh and valuable insight into the relationship between democracy and corruption, and that many more opportunities lie ahead. For example, in our model, agents were placed in a social network such that network links represented friendships mediating social influence and peer pressure. One could examine many other aspects of how friendships affect a pair, or a small group, of agent friends. Evolutionary game theory is ideal for doing so because of the versatile adaptive mechanisms included in its foundations [45]: (i) agents can act strategically, (ii) strategies can spread via learning, (iii) strategic interactions can precipitate cooperation among unrelated individuals, (iv) effective cooperation can give birth to social norms, and (v) social norms can explain apparent irrational behaviours. With this in mind, we may expect that incorporating the elements of evolutionary game theory into our framework to examine conditions under which parochialists dominate altruists and vice versa will ultimately lead to rich dynamics [46] and a plethora of surprising discoveries. We hope that such findings will materialize sooner rather than later, given the role of democracy in the modern world and the strain that corruption places on efforts to make this world a better place.

Placing our findings in the general context of complex-system dynamics, there are many real-world systems in which the natural deterioration of the system's constituents via faults, damage or malignancies causes an often sudden transition to a dysfunctional mode of operation [47,48]. Innate controls (e.g. the immune system) or human interventions (e.g. risk management in finance, ecosystem conservation in ecology or disease prevention in medicine) are designed to counteract the disruptive forces and boost the system's resilience. The ensuing race between the negative effects of deterioration and the positive effects of control or intervention are likely to determine the system's longevity and ultimate fate. This is the case in the present study of human society, but also in cellular collectives behind dementia in neurobiology [49] or diabetes in physiology [50]. Understanding that the same dynamical principles govern systems that at a glance seem widely different offers a unifying perspective across disciplines by which effective solutions from one domain may inspire effective solutions in very different domains.

## 5. Methods

We conducted the present study in three stages: data collection, empirical analyses and mathematical modelling. A description of each stage follows hereafter.

To investigate the relationship between democracy and corruption, we relied on well-documented indices published by independent organizations. To characterize democracy we used the Index of Democracy by the Economist Intelligence Unit [51]. This index relies on

60 indicators, grouped into five categories (electoral process and pluralism, civil liberties, the functioning of government, political participation and political culture). For each category a rating on a 0–10 scale is deduced, upon which the overall index is obtained as a simple average of the five categories. As an alternative, we used the Global Freedom Score by Freedom House [52]. Although the name suggests a focus on freedom rather than democracy, the scores are based on categories that overlap with those that constitute the Index of Democracy. There are 10 political rights indicators divided into three categories (electoral process, political pluralism and participation, and functioning of government) and 15 civil liberties indicators divided into four categories (freedom of expression and belief, associational and organizational rights, rule of law, and personal autonomy and individual rights). The highest political-rights score is 40 and civil-liberties score is 60. These two scores are finally summed to obtain the overall score for a given country.

To characterize corruption, we used the CPI by Transparency International [53]. This is a composite index based on 13 data sources from 12 institutions that measure perceptions of corruption over the past 2 years. All data are standardized to a scale between 0 and 100, upon which the overall score for a country is calculated as the average of all standardized scores available. Scores are rounded to whole numbers. Because a higher score indicates less corruption, the index can be interpreted as measuring how well a country fights or suppresses corruption. As an alternative, we used the Control of Corruption by the World Bank [54]. This indicator (together with five related indicators published by the World Bank) is based on several hundred variables obtained from 31 different data sources. The goal is to capture perceptions of how much public power is exercised for private gain, encompassing from petty to grand corruption. The aggregate result for a given country is obtained via a statistical tool named the unobserved components model [54], producing the values with zero mean, unit standard deviation and a range approximately between $-2.5$ and $2.5$.

We supplemented the above-stated data with the average voter turnout between 2009 and 2019 for a subset of 50 countries, including all 27 EU members. For each of these countries, we also extracted the nominal GDP per capita in 2019 as reported by the World Bank along with the latest information on countries' population size. All data used in this study are available for download at doi:10.17605/OSF.IO/UGE4V.

Hypothesizing that the relationship between democracy and corruption is nonlinear, we plotted the CPI against the Index of Democracy, and fitted the data using a piecewise-linear model

$$y = \begin{cases} a_1 x + b & x < x_c, \\ a_2 (x - x_c) + a_1 x_c + b & x \geq x_c, \end{cases} \tag{5.1}$$

where $a_1$, $a_2$, $b$ and $x_c$ are the model coefficients. We estimated these coefficients and the corresponding 95% confidence intervals using the Levenberg–Marquardt algorithm [55]. We repeated the same estimation procedure on an independent dataset, specifically the Control of Corruption against the Global Freedom Score.

We further hypothesized that voter turnout may depend on a number of factors, in particular corruption suppression, democratic norms, gross domestic product *per capita*, membership of the EU and population size. We therefore fitted the data with a model of the form

$$\mathbf{y} = \beta_1 \mathbf{x}_1 + \beta_2 \mathbf{x}_2 + \beta_3 \mathbf{x}_3 + \beta_4 \mathbf{x}_4 + \beta_5 \mathbf{x}_5, \tag{5.2}$$

where $\mathbf{y}$ is a column vector comprising voter turnout as a dependent variable, $\beta_i$ are the model coefficients and $\mathbf{x}_i$ are column vectors corresponding to the five explanatory variables listed above. Because economic data often suffer from multicollinearity, we estimated the model coefficients using the principal component regression [56]. This method consists of the following steps. First, the data matrix $\mathbf{X} = [\mathbf{x}_1, \ldots, \mathbf{x}_5]$ is centred and in our case normalized. Second, matrix factorization $\mathbf{X}^T \mathbf{X} = \mathbf{V} \mathbf{\Lambda} \mathbf{V}^T$ is performed, where the columns of $\mathbf{V}$ constitute the orthonormal set of eigenvectors of $\mathbf{X}^T \mathbf{X}$ and the diagonal of $\mathbf{\Lambda}$ contains the corresponding non-negative eigenvalues of $\mathbf{X}^T \mathbf{X}$ in descending order. Third, a new data matrix $\mathbf{W}_k = [\mathbf{X} \mathbf{v}_1, \ldots, \mathbf{X} \mathbf{v}_k]$, $1 \leq k \leq 5$, is formed

using the columns $\mathbf{v}_j$ of $\mathbf{V}$. Fourth, a vector of coefficients $\boldsymbol{\gamma}_k = (\mathbf{W}_k^T \mathbf{W}_k)^{-1} \mathbf{W}_k^T \mathbf{y}$ is calculated in order to finally obtain a vector of original model coefficients $\boldsymbol{\beta}_k = \mathbf{V}_k \boldsymbol{\gamma}_k$, where $\mathbf{V}_k = [\mathbf{v}_1, \ldots, \mathbf{v}_k]$. Finally, because there are five possible models, $1 \leq k \leq 5$, the best one must be selected, which we did by relying simply on the adjusted coefficient of determination. The key advantage of principal component regression is that restricting the number of vectors $\mathbf{v}_j$ in constructing the matrix $\mathbf{W}_k$ to $1 \leq k < 5$ reduces the variance of the estimator $\hat{\boldsymbol{\beta}}_k$ compared with the full model estimator $\hat{\boldsymbol{\beta}}_5$ without losing much in terms of the goodness of fit.

Implementation of the mathematical-modelling aspects of this study followed model assumptions listed in the main text. We used the Julia programming language to run numerical simulations. Sample code is available for download at doi:10.17605/OSF.IO/UGE4V.

Data accessibility. All data and sample code used herein are available for download at doi:10.17605/OSF.IO/UGE4V.

Authors' contributions. B.P., M.J. and T.L. conceived the research. D.K. and M.J. performed simulations. B.P. and T.L. obtained and analysed the data. All coauthors discussed the results and wrote the manuscript. B.P. and M.J. contributed equally.

Competing interests. The authors declare no conflicts of interests.

Funding. B.P. was supported by the Slovenian Research Agency (grant no. J7-3156). B.P. and T.L. were supported by the Centre of Excellence project 'DATACROSS' no. KK.01.1.1.01.0009, co-financed by the Croatian Government and the European Union through the European Regional Development Fund— the Competitiveness and Cohesion Operational Programme. M.J. was supported by JSPS KAKENHI grant nos. JP20H04288 and JP21K04545. D.K. was supported by the Slovenian Research Agency, research core funding programme no. P3-0396 and project nos. N3-0048, N3-0133 and J3-9289. P.H. was supported by JSPS KAKENHI grant no. JP21H04595. JSPS stands for the Japan Society for the Promotion of Science.

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
