## [Peer Review File · Proceedings. Mathematical, Physical, and Engineering Sciences]

Review History

RSPA-2021-0567.R0 (Original submission)

Review form: Referee 1

Is the manuscript an original and important contribution to its field?

Good

Is the paper of sufficient general interest?

Good

Is the overall quality of the paper suitable?

Good

Can the paper be shortened without overall detriment to the main message?

Yes

Do you think some of the material would be more appropriate as an electronic appendix?

No

Do you have any ethical concerns with this paper?

No

Recommendation?

Accept with minor revision (please list in comments)

Comments to the Author(s)

This manuscript presents research about a real-data analysis and an agent-based model on the relationship between democracy and corruption.

I like the idea; furthermore, the approach, as far as I know, is original. However, I strongly recommend that the authors take the following comments into account:

The study consists of two parts. In the first part, the authors study the correlation between democracy and corruption for different countries through four data sets arranged by pairs. I am not sure why they choose those four data sets and why they arrange them in that way (i.e., A Vs B and C Vs D but not A Vs C and B Vs D, for example). Perhaps the authors should justify their choices.

Visual inspection of data shows a positive correlation between democracy and corruption control and even the tipping point. Besides that, I wonder if the authors have checked other fits besides Eq. 7. Why do they use a piece-wise linear model? Does it fit better than other continuous (and relatively simple) functions? Besides that point, I appreciate the mathematical rigor. For example, principal component regression seems to be an appropriate method for the model's coefficient estimation.

Regarding the second part of the research, the model looks elaborated and sensible. Although it has many components, the mean-field approach helps to understand the influence of some of them on the dynamics. The numerical results are interesting, showing non-trivial dynamics and a tipping point.

My main concern refers to the connection between both parts of the study, i.e., the justification of the model. If I understand the manuscript, this connection relies on the interpretation of the so-called "parochial ingroup bias" in terms of corruption. I miss a discussion about this point and, if possible, some references to the corresponding sociological theories (a field that, honestly, I do not know in depth.).

Review form: Referee 2

Is the manuscript an original and important contribution to its field?

Marginal

Is the paper of sufficient general interest?

Acceptable

Is the overall quality of the paper suitable?

Acceptable

Can the paper be shortened without overall detriment to the main message?

Yes

Do you think some of the material would be more appropriate as an electronic appendix?

No

Do you have any ethical concerns with this paper?

No

Recommendation?

Major revision is needed (please make suggestions in comments)

Comments to the Author(s)

This paper presents evidence of a bi-modal relationship between level of democracy and corruption suppression in countries. It then constructs a mechanistic model which shows the same behavior. As I am a physicist and not very familiar with the social science literature on this, my comments mostly pertain the modeling part.

I think the model is interesting and merits consideration. But at the same time, I find many of the assumptions not well justified and based on anecdotal evidence. Thus, I recommend some revisions before the paper is published. In summary, my main issue is that many assumptions going into the model are based on personal presumptions about the behavior of societies, rather than well established results of social studies.

Details below.

1. You state variables (i.e. 2 or 3 spin states) should be such that they really specify the state of the agent. However, you don't have a state representation for the "Parochialists" you just say they are "recruited from the oportunists" where their beliefs are cemented. This is either saying that the spin rep is not a good way to represent the states, or you should have a spin state for the parochial state.
2. Assumptions 4-6 seem quite anecdotal and debatable. Can you either add citations or discuss alternative scenarios? For example, #6 assuming democratic majority wants to curb parochial behavior, is quite a strong assumption. In history, it seems that societies took a long time to learn such behaviors. And it is quite possible that many societies do not exhibit such behavior, either because of other priorities, such as economy or other hardships.
3. equation (3) seems quite arbitrary and a complex choice for a basic assumption of the model.
4. Can you conduct, or do you already have, ablation studies in which you check the effect of each of the assumptions? For example, what happens if 4-6 are removed? It is clear that once you add sufficiently many parameters to a model you can fit arbitrarily complex behavior.

Decision letter (RSPA-2021-0567.R0)

06-Oct-2021

Dear Mr Lipic

The Editor of Proceedings A has now received comments from referees on the above paper and would like you to revise it in accordance with their suggestions which can be found below (not including confidential reports to the Editor).

Please submit a copy of your revised paper within four weeks - if we do not hear from you within this time then it will be assumed that the paper has been withdrawn. In exceptional circumstances, extensions may be possible if agreed with the Editorial Office in advance.

Please note that it is the editorial policy of Proceedings A to offer authors one round of revision in which to address changes requested by referees. If the revisions are not considered satisfactory by the Editor, then the paper will be rejected, and not considered further for publication by the journal. In the event that the author chooses not to address a referee's comments, and no scientific justification is included in their cover letter for this omission, it is at the discretion of the Editor whether to continue considering the manuscript.

To revise your manuscript, log into <http://mc.manuscriptcentral.com/prsa> and enter your Author Centre, where you will find your manuscript title listed under "Manuscripts with Decisions." Under "Actions," click on "Create a Revision." Your manuscript number has been appended to denote a revision.

You will be unable to make your revisions on the originally submitted version of the manuscript. Instead, revise your manuscript and upload a new version through your Author Centre.

When submitting your revised manuscript, you will be able to respond to the comments made by the referee(s) and upload a file "Response to Referees" in Step 1: "View and Respond to Decision Letter". Please provide a point-by-point response to the comments raised by the reviewers and the editor(s). A thorough response to these points will help us to assess your revision quickly. You can also upload a 'tracked changes' version either as part of the 'Response to reviews' or as a 'Main document'.

IMPORTANT: Your original files are available to you when you upload your revised manuscript. Please delete any unnecessary previous files before uploading your revised version.

When revising your paper please ensure that it remains under 28 pages long. In addition, any pages over 20 will be subject to a charge (£150 + VAT (where applicable) per page). Your paper has been ESTIMATED to be 13 pages.

Open Access

You are invited to opt for open access, our author pays publishing model. Payment of open access fees will enable your article to be made freely available via the Royal Society website as soon as it is ready for publication. For more information about open access please visit <https://royalsociety.org/journals/authors/open-access/>. The open access fee for this journal is £1700/\$2380/€2040 per article. VAT will be charged where applicable. Please note that if the corresponding author is at an institution that is part of a Read and Publishing deal you are required to select this option. See <https://royalsociety.org/journals/librarians/purchasing/read-and-publish/read-publish-agreements/> for further details.

Once again, thank you for submitting your manuscript to Proc. R. Soc. A and I look forward to receiving your revision. If you have any questions at all, please do not hesitate to get in touch.

Yours sincerely
Raminder Shergill
proceedingsa@royalsociety.org

on behalf of
Professor Matjaz Perc
Board Member
Proceedings A

Reviewer(s)' Comments to Author:

Referee: 1

Comments to the Author(s)

This manuscript presents research about a real-data analysis and an agent-based model on the relationship between democracy and corruption.

I like the idea; furthermore, the approach, as far as I know, is original. However, I strongly recommend that the authors take the following comments into account:

The study consists of two parts. In the first part, the authors study the correlation between democracy and corruption for different countries through four data sets arranged by pairs. I am not sure why they choose those four data sets and why they arrange them in that way (i.e., A Vs B and C Vs D but not A Vs C and B Vs D, for example). Perhaps the authors should justify their choices.

Visual inspection of data shows a positive correlation between democracy and corruption control and even the tipping point. Besides that, I wonder if the authors have checked other fits besides Eq. 7. Why do they use a piece-wise linear model? Does it fit better than other continuous (and relatively simple) functions? Besides that point, I appreciate the mathematical rigor. For example, principal component regression seems to be an appropriate method for the model's coefficient estimation.

Regarding the second part of the research, the model looks elaborated and sensible. Although it has many components, the mean-field approach helps to understand the influence of some of them on the dynamics. The numerical results are interesting, showing non-trivial dynamics and a tipping point.

My main concern refers to the connection between both parts of the study, i.e., the justification of the model. If I understand the manuscript, this connection relies on the interpretation of the so-called "parochial ingroup bias" in terms of corruption. I miss a discussion about this point and, if possible, some references to the corresponding sociological theories (a field that, honestly, I do not know in depth.).

Referee: 2

Comments to the Author(s)

This paper presents evidence of a bi-modal relationship between level of democracy and corruption suppression in countries. It then constructs a mechanistic model which shows the same behavior. As I am a physicist and not very familiar with the social science literature on this, my comments mostly pertain the modeling part.

I think the model is interesting and merits consideration. But at the same time, I find many of the assumptions not well justified and based on anecdotal evidence. Thus, I recommend some revisions before the paper is published. In summary, my main issue is that many assumptions going into the model are based on personal presumptions about the behavior of societies, rather than well established results of social studies.

Details below.

1. You state variables (i.e. 2 or 3 spin states) should be such that they really specify the state of the agent. However, you don't have a state representation for the "Parochialists" you just say they are "recruited from the oportunists" where their beliefs are cemented. This is either saying that the spin rep is not a good way to represent the states, or you should have a spin state for the parochial state.
2. Assumptions 4-6 seem quite anecdotal and debatable. Can you either add citations or discuss alternative scenarios? For example, #6 assuming democratic majority wants to curb parochial behavior, is quite a strong assumption. In history, it seems that societies took a long time to learn such behaviors. And it is quite possible that many societies do not exhibit such behavior, either because of other priorities, such as economy or other hardships.
3. equation (3) seems quite arbitrary and a complex choice for a basic assumption of the model.
4. Can you conduct, or do you already have, ablation studies in which you check the effect of each of the assumptions? For example, what happens if 4-6 are removed? It is clear that once you add sufficiently many parameters to a model you can fit arbitrarily complex behavior.

Board Member:

Comments to Author(s):

We will be happy to consider a revised manuscript that takes the comments of both Referees into account.

RSPA-2021-0567.R1 (Revision)

Review form: Referee 1

Is the manuscript an original and important contribution to its field?

Good

Is the paper of sufficient general interest?

Good

Is the overall quality of the paper suitable?

Good

Can the paper be shortened without overall detriment to the main message?

Yes

Do you think some of the material would be more appropriate as an electronic appendix?

No

Do you have any ethical concerns with this paper?

No

Recommendation?

Accept as is

Comments to the Author(s)

I appreciate the explanations and clarifications by the authors.

In my opinion, all the referees' concerns have been satisfactorily answered.

Therefore, I recommend accepting the manuscript for publication in SRPA Journal.

Review form: Referee 2

Is the manuscript an original and important contribution to its field?

Acceptable

Is the paper of sufficient general interest?

Good

Is the overall quality of the paper suitable?

Good

Can the paper be shortened without overall detriment to the main message?

Yes

Do you think some of the material would be more appropriate as an electronic appendix?

No

Do you have any ethical concerns with this paper?

No

Recommendation?

Accept as is

Comments to the Author(s)

I am satisfied with the response and have no further questions.

Thank you for pointing me to the appropriate references and supplemental figures. I find the ablation studies adequate and helpful in understanding the effects of your assumptions. Although I am not an expert in the social science side of the matter, the references you mentioned should suffice to justify the assumptions.

Hence, only assumption 6 seems to be your choice. That is fine, as long as you clearly state this, saying that with assumption 6 you are able to best reproduce the regression discontinuity observed in the data.

Decision letter (RSPA-2021-0567.R1)

01-Dec-2021

Dear Mr Lipic

I am pleased to inform you that your manuscript entitled "The microdynamics shaping the relation between democracy and corruption" has been accepted in its final form for publication in Proceedings A.

Our Production Office will be in contact with you in due course. You can expect to receive a proof of your article soon. Please contact the office to let us know if you are likely to be away from e-mail in the near future. If you do not notify us and comments are not received within 5 days of sending the proof, we may publish the paper as it stands.

As a reminder, you have provided the following 'Data accessibility statement' (if applicable). Please remember to make any data sets live prior to publication, and update any links as needed when you receive a proof to check. It is good practice to also add data sets to your reference list. *Statement (if applicable):* All data and sample code used in this work are made available for download at doi.org/10.17605/OSF.IO/UGE4V

Under the terms of our licence to publish you may post the author generated postprint (ie. your accepted version not the final typeset version) of your manuscript at any time and this can be made freely available. Postprints can be deposited on a personal or institutional website, or a recognised server/repository. Please note however, that the reporting of postprints is subject to a media embargo, and that the status the manuscript should be made clear. Upon publication of the definitive version on the publisher's site, full details and a link should be added.

You can cite the article in advance of publication using its DOI. The DOI will take the form: 10.1098/rspa.XXXX.YYYY, where XXXX and YYYY are the last 8 digits of your manuscript number (eg. if your manuscript number is RSPA-2017-1234 the DOI would be 10.1098/rspa.2017.1234).

For tips on promoting your accepted paper see our blog post:
<https://royalsociety.org/blog/2020/07/promoting-your-latest-paper-and-tracking-your-results/>

On behalf of the Editor of Proceedings A, we look forward to your continued contributions to the Journal.

Sincerely,
Raminder Shergill
proceedingsa@royalsociety.org

on behalf of
Professor Matjaz Perc
Board Member
Proceedings A

Reviewer(s)' Comments to Author:

Referee: 1

Comments to the Author(s)

I appreciate the explanations and clarifications by the authors.

In my opinion, all the referees' concerns have been satisfactorily answered.

Therefore, I recommend accepting the manuscript for publication in SRPA Journal.

Referee: 2

Comments to the Author(s)

I am satisfied with the response and have no further questions.

Thank you for pointing me to the appropriate references and supplemental figures. I find the ablation studies adequate and helpful in understanding the effects of your assumptions. Although I am not an expert in the social science side of the matter, the references you mentioned should suffice to justify the assumptions.

Hence, only assumption 6 seems to be your choice. That is fine, as long as you clearly state this, saying that with assumption 6 you are able to best reproduce the regression discontinuity observed in the data.